# The Benefits, Challenges, and Strategies toward Establishing a Community-Engaged Knowledge Hub: An Integrative Review

**DOI:** 10.3390/ijerph20021160

**Published:** 2023-01-09

**Authors:** Jasleen Brar, Nashit Chowdhury, Mohammad M. H. Raihan, Ayisha Khalid, Mary Grantham O’Brien, Christine A. Walsh, Tanvir C. Turin

**Affiliations:** 1Department of Family Medicine, Cumming School of Medicine, University of Calgary, Calgary, AB T2N 4N1, Canada; 2Newcomer Research Network, University of Calgary, Calgary, AB T2N 1N4, Canada; 3Department of Community Health Sciences, Cumming School of Medicine, University of Calgary, Calgary, AB T2N 4N1, Canada; 4School of Languages, Linguistics, Literatures and Cultures, University of Calgary, Calgary, AB T2N 1N4, Canada; 5Faculty of Social Work, University of Calgary, Calgary, AB T2N 1N4, Canada

**Keywords:** knowledge creation, knowledge mobilization, hub, community engagement, community involvement, transdisciplinary collaboration, cross-sectorial partnership

## Abstract

Current knowledge creation and mobilization efforts are concentrated in academic institutions. A community-engaged knowledge hub (CEKH) has the potential for transdisciplinary and cross-sectorial collaboration between knowledge producers, mobilizers, and users to develop more relevant and effective research practices as well as to increase community capacity in terms of knowledge production. **Objective:** To summarize existing original research articles on knowledge hubs or platforms and to identify the benefits, challenges, and ways to address challenges when developing a CEKH. **Methods:** This study followed a systematic integrative review design. Following a comprehensive search of academic and grey literature databases, we screened 9030 unique articles using predetermined inclusion criteria and identified 20 studies for the final synthesis. We employed thematic analysis to summarize the results. **Results:** The focus of the majority of these knowledge mobilization hubs was related to health and wellness. Knowledge hubs have a multitude of benefits for the key stakeholders including academics, communities, service providers, and policymakers, including improving dissemination processes, providing more effective community interventions, ensuring informed care, and creating policy assessment tools. Challenges in creating knowledge hubs are generally consistent for all stakeholders, rather than for individual stakeholders, and typically pertain to funding, resources, and conflicting perspectives. As such, strategies to address challenges are also emphasized and should be executed in unison. **Conclusions:** This study informs the development of a future CEKH through the identification of the benefits, challenges, and strategies to mitigate challenges when developing knowledge hubs. This study addresses a literature gap regarding the comparisons of knowledge hubs and stakeholder experiences.

## 1. Background

Newly created knowledge is typically limited to knowledge producers and the immediate silos that surround knowledge producers [1]. In many instances, knowledge mobilization and translation efforts remain concentrated within the confinement of academic venues and channels—creating barriers between knowledge producers and knowledge users [2]. Albeit a rapidly globalizing and interconnecting world, community-level knowledge users seldom collaborate in the creation and dissemination of knowledge [1].

Knowledge is understood as a culmination of three processes. Knowledge production refers to the creation of new knowledge based on experimentation and/or summarization of old knowledge to create new meanings [3]. Knowledge mobilization refers to the process by which knowledge is made accessible and better understood by a wide variety of audiences [3]. Knowledge translation refers to the application of knowledge, and in the context of healthcare, for example, to improve health, provide more effective health services, and strengthen the healthcare system [4].

Within these processes, knowledge production is most often prioritized, with mobilization and translation efforts largely limited to academics and knowledge-producing institutions [1,2]. However, knowledge mobilization and translation processes are equally important to increase the dissemination of emerging knowledge and raise awareness of knowledge production processes [2]. In fact, knowledge mobilization and translation efforts make knowledge production more meaningful [3]. Exposure to knowledge production processes is also necessary to promote the involvement of external groups, such as community members who are impacted by emerging knowledge, in knowledge production processes [3,5]. Furthermore, removing barriers between academia and communities allows for the perspectives of community members to be better integrated into knowledge production processes and allow knowledge production teams to better reflect the diverse communities of knowledge users [5].

Various stakeholders should collaborate in all knowledge processes [6,7,8]. Knowledge production, mobilization, and translation efforts require a transdisciplinary and cross-sectorial approach to increase the impact of knowledge production and increase connectivity between academic institutions and the larger society [9]. Academics, community members, service providers, and policymakers are all important stakeholders in knowledge production, mobilization, and translation processes [6,7,8]. It is crucial for all stakeholders to have the ability to connect and collaborate in order to effectively execute knowledge processes and increase the widespread uptake of emerging knowledge and research.

To promote transdisciplinary and cross-sectorial community-focused research and knowledge mobilization, the Newcomer Research Network (https://www.ucalgary.ca/network/newcomer-research/home) was established to advance and advocate for research that will provide information about intercultural practices and support newcomers in Calgary and the surrounding communities, nationally and internationally [10]. The authors of this review belong to this research network. The network organized two knowledge mobilization symposia (2019 and 2021) that brought stakeholders from various disciplines and sectors together, where all voiced the need for the development of a community-engaged knowledge hub (CEKH) for the research involving diverse communities. A policy brief was published based on these activities that argued the need for the development of such knowledge hubs [11]. We also presented the recommendations received from the participants and identified key focus areas that may work for the development of a CEKH for the diverse communities. We, collectively, recognized the need of building collaborative research and knowledge mobilization capacity with diverse communities, which brings together a range of actors and stakeholders such as academic researchers, think tanks, governmental and nongovernmental policy makers, service provider organizations, and newcomer grassroots community organizations [10,11].

CEKHs provide such opportunities for vast stakeholder collaboration in all aspects of the research and knowledge dissemination process. Knowledge hubs are typically physical spaces or virtual platforms that allow for interdisciplinary discussions and the provision of resources to increase the capacity of research teams and improve knowledge production processes [12]. A CEKH will have equitable and empowered involvement of the community members to ensure impactful knowledge creation and mobilization. A CEKH can also be used for workshops to educate communities on emerging knowledge—thus increasing knowledge mobilization. The involvement of community members in knowledge production, as encouraged by collaborative spaces of knowledge hubs, also increases knowledge mobilization [13]. Moreover, the involvement of policymakers and service providers in knowledge hubs and collaborative members of the knowledge production process allows for enhanced and more effective translation of knowledge directly into policy and practices [14,15].

Globally, the literature on CEKHs or knowledge hubs at large is not extensive. Although much of the research includes the benefits and challenges presented by individual hubs, there is an apparent lack of encompassing literature reviews that draw from these studies. Furthermore, there is a paucity of research identifying commonalities between hubs and the resultant lessons/strategies for new knowledge hub developers as well as a lack of studies that highlight the connections to and implications for the community. One of the main purposes of knowledge hubs can and should be community collaboration to produce better outcomes for the community [16]. Without the identification of benefits for community members and other stakeholders such as service providers and policymakers, knowledge hubs are only partially understood in terms of their invaluable potential. Knowledge hubs have been advanced as an exemplary tool for community engagement and transdisciplinary collaboration throughout all knowledge-related processes [13,14,15,16].

The objectives of this study are to analyze and summarize existing literature on various kinds of knowledge hubs that collaborated with the community in some way. This systematic integrative review draws on the existing theoretical and empirical studies and describes the benefits, challenges, and ways to mitigate challenges for the development of a wide range of knowledge hubs.

## 2. Methods

### 2.1. Research Design

We conducted a systematic integrative review of existing partnered and/or organizational approaches for knowledge mobilization activities through knowledge hubs. An integrative review is a specific review method that summarizes data from previous studies of diverse methodologies with the goal of providing a more comprehensive understanding of a particular issue in order to contribute to theory development and inform practice and policy [17,18]. This study follows the steps (Figure 1) of an integrative literature review as identified by existing literature: (1) problem identification, (2) literature search, (2) data collection, (3) data evaluation, (4) data analysis, and (5) interpretation and presentation of results [17,18].

### 2.2. Problem Identification

As identified by our study objectives, the problem—within the scope of research on knowledge hubs—is the lack of literature reviews that compare and summarize different hubs, albeit the abundant literature on individual knowledge hubs. Thus, it is necessary to understand knowledge hubs and related research through the lens of a literature review to gain a better understanding of overlapping benefits and challenges presented in the development of knowledge hubs, as well as how hub stakeholders have addressed such challenges.

### 2.3. Literature Search

To identify relevant studies related to our study objectives, we conducted a comprehensive systematic search of published and grey literature through academic and grey literature databases (Table 1). We used several keywords synonymous with or related to knowledge mobilization and knowledge hubs to identify appropriate studies (Table 2). We also combined some of the main keywords to form search strings to use in the MEDLINE and Embase databases (Appendix A).

Initially, we identified 11,803 papers from academic databases and 2227 records from grey literature sources, based on the above search strategies (Figure 1). The online program COVIDENCE was used to screen all records during the data evaluation stage. After duplicate paper removal, we had 9030 papers. The details of the articles derived from academic and grey literature search are provided in an online data repository, figshare [19,20]. Two reviewers, JB and AK, independently screened the titles and abstracts of the 9030 records and excluded 8604 papers based on predetermined criteria (Table 3). Differences in exclusion were resolved through discussion with a third reviewer (TCT). During the title and abstract screening, papers that were not in English, those that focused on knowledge management instead of mobilization, or those that focused on evidence-based practices and/or knowledge translation within a single group of stakeholders (rather than collaboration with multiple stakeholders) were excluded (Table 3).

The remaining 426 papers underwent a full-text review for eligibility by JB. During the full-text review, articles on online platforms/networks, articles describing or evaluating existing hubs (rather than developing new hubs), and articles without community connections or stakeholders were excluded, so that 20 studies remained for the complete qualitative analysis. We used a PRISMA flow chart to track the number of studies following each stage of exclusion (Figure 2) [21].

### 2.4. Data Evaluation

For an integrative review, which includes various study designs from diverse empirical sources, evaluation of the studies can be very complicated [17,18]. The data evaluation for integrative reviews depends on the sampling frame [17,18]. Whittemore and Knafl suggested that, for an integrative review, which includes studies with multiple study designs, it is reasonable to evaluate the quality of studies that present outlier findings to examine if that findings could be due to methodological quality [18]. In our study, we included diverse methodological papers and focused on a range of outcomes for our reporting. During our analysis process, we did not identify any studies with outlier findings that could be deemed to have resulted by methodological discrepancy.

### 2.5. Data Analysis

The final sample of 20 studies was analyzed in NVivo 12. We applied a qualitative description approach to code and thematically analyze the studies [22]. We used an open coding method, in line with the qualitative description approach [22,23]. We looked for benefits, challenges, and ways to address challenges by referring to our study objectives, through a ‘line by line’ coding manner and coded any phrase, sentence, or paragraph that referred to one or more of the three [22]. The code was named to best capture the phrase, sentence, or paragraph in a few words that could be thematically grouped [24].

### 2.6. Presentation

Each of the three sets of codes—benefits, challenges, and ways to address challenges—was grouped under one or more of the primary stakeholders we have identified as part of knowledge hub teams: academics, community members, service providers, and policymakers (Figure 2) [24]. Certain codes that included benefits for multiple stakeholders or challenges for multiple stakeholders, for example, were categorically included under all stakeholders the code pertained, and thus, some codes were categorically repeated. Codes pertaining to each stakeholder that were related and/or able to be grouped together were categorized under single themes.

## 3. Results

### 3.1. Study Characteristics

Table 4 shows that most of the studies (13/20) were conducted in North America, among which seven studies were in the USA and six were in Canada. Most other knowledge hubs were in European countries, including Germany (*n* = 2; two studies about the same knowledge hub), the UK (*n* = 2), Netherlands (*n* = 2), and only one was in multiple countries namely Jamaica, Kenya, Uganda, and South Africa. Most of these studies used a qualitative (*n* = 16) descriptive approach to discuss a particular knowledge hub. Among them, 13 were case studies, two were content analyses, and one was a narrative review. Three studies were quantitative including one quasi-experimental study, one survey, and one that was an analysis of archival and documentary data. The only mixed study summarized the results from multiple randomized controlled trials and described how they were done under a knowledge hub. Three studies used a community-based participatory approach during the building and/or activities of the hubs. Eight studies mentioned knowledge translation/mobilization/exchange/transfer principles that guided their work and the creation of the knowledge hubs. One study followed implementation research guidelines and one based its hub on community engagement. The focus of the majority of these knowledge hubs (*n* = 15) was related to health and wellness. These include seniors’ health [25], bipolar disorder [26], occupational health [27,28], schizophrenia [29,30], autism spectrum disorders [31], mental health [32,33], kidney diseases [34], HIV [35], healthcare in general [6,36], nursing [37], and public health systems [38]. Other knowledge hubs were on education [39], social work [40], sustainability [41], creative economy [42], and science in general, particularly open science concept [43]. In terms of initiation of the knowledge hubs, most of them were initiated mainly by researchers (*n* = 8): three were through a collaborative effort between researchers, policymakers and/or community; two were by policymakers; and one was by service providers. Figure 3 shows the different collaborating sectors of the knowledge hub.

### 3.2. Benefits

Sixty-eight codes and 17 themes were identified and categorized as benefits for developing knowledge hubs (Table 5). A total of 21 codes were classified as community benefits, 18 codes for academics, 5 codes for service providers and policymakers each, and 19 codes as benefits for all stakeholders. The most common codes are ‘knowledge production’ (*n* = 20), ‘knowledge mobilization’ (*n* = 19), ‘community-need sensitive interventions’ (*n* = 18), ‘knowledge exchange’ (*n* = 16), and ‘improved community care’ (*n* = 14). The most common themes were ‘research’ benefits for academics, ‘appropriate community interventions’ for communities, ‘improved care provision’ for service providers, and ‘knowledge translation’ for policymakers. Two of the most frequently coded benefits were directly for the community, and the other three knowledge-related processes stemmed from knowledge hubs that collaborated with community members.

### 3.3. Challenges

Thirty-seven codes and 16 themes were identified as challenges when developing knowledge hubs (Table 5). Seven codes were classified as challenges for/by community members, 5 challenges for/by academics, 2 for/by policymakers, 23 for/by all stakeholders, and 3 challenges were reported from or due to the collaboration with service providers. The most frequently coded challenges are ‘limited funding’ (*n* = 9), ‘time-consuming’ (*n* = 6), and ‘conflicts establishing in the healthcare system’ (*n* = 6). The most common themes are ‘challenges with new ideas’ for academics and ‘poor trust in research/academia’ for communities; ‘conflicts with external systems’ is the only theme pertaining to policymakers. Challenges related to ‘starting up a new knowledge hub’ were the most frequent theme for all stakeholders.

### 3.4. Addressing the Challenges

Sixteen codes and four themes were identified for strategies/advice in studies related to addressing the challenges presented during knowledge hub development (Table 6). Due to the limited amount of literature that described ways to address challenges as well as the emphasis on unified execution by all stakeholders when the ways to address challenges were described, these codes and themes were not individually categorized into the four stakeholder categories, as the benefits and challenges of developing knowledge hubs were.

## 4. Discussion

This study reveals the immense benefits of knowledge hubs for all stakeholders involved. Specifically, as we focused on original studies where the researchers collaborated with the community to build knowledge hubs, we were able to identify benefits specific to collaborating with communities in knowledge hubs, thereby addressing the current literature gap in a CEKH. Furthermore, the majority of benefits, with some of the most frequent codes, were for communities in comparison to other stakeholders. The literature review supports the clear advantages of collaboration with the communities in knowledge hubs including improved health outcomes, capacity building, and greater involvement in knowledge production and mobilization activities.

This study also highlights benefits for academics and research teams working in conjunction with communities through knowledge hubs. These stakeholders are better able to conduct relevant research that can promote appropriate and more useful community interventions, pull on a greater population of research participants, and gain greater exposure to perspectives of actual knowledge users—rather than only the perspectives of other knowledge producers and/or mobilizers within academic institutions [38]. Another key benefit for academics, through knowledge hubs, is the improvement of the academic image [28]. Through the inclusion of and collaboration with community members, service providers, and policymakers, studies in this review revealed that the perceptions of these three stakeholders regarding academia improved and shifted from the common separatist and elitist view of academia to one of collegiality and participatory learning [28,36,37].

Albeit comparatively less stated in the literature, service providers and policymakers also attain benefits from involvement in collaborative knowledge hubs. Service providers, such as physicians and other care providers, benefit from the collaborative processes of knowledge hubs, as they are able to directly work with those who produce knowledge (academics) and those who receive knowledge from service providers (communities). Therefore, service providers, according to this body of research, can increase their capacity to provide better care, as well as be involved in research processes themselves to encourage and work on research related to their field of work [25,34,35,40]. Furthermore, a number of benefits accrue to policymakers in terms of knowledge translation, such as being able to access more research that can better inform policies, as well as in terms of policy assessment through interactions with the larger public, who are affected by such policies, through collaborative spaces in knowledge hubs [6,31,34]. There are also numerous benefits for all stakeholders collaborating in knowledge hubs. As knowledge hubs are a unifying space, all stakeholders are able to expand their networks and increase capacity through knowledge exchange [6,39]. They can gain better access to resources for—including but not limited to—knowledge production, mobilization, and/or translation purposes [28,37].

As indicated by the papers and our analysis, knowledge hubs present many diverse benefits for all collaborating stakeholders. Nonetheless, certain challenges exist. From our analysis, challenges in the development of knowledge hubs arise for academics, community members, and policymakers, as well as for all stakeholders. Although no challenges arose specific to service providers, challenges that pertain to all stakeholders are assumed to include service providers. The increased number of challenges for all stakeholders, in comparison to the number of individual challenges for academics, community members, and policymakers, could result from the unified nature of knowledge hubs that allow for unified challenges to be experienced by all stakeholders. A challenge experienced by one group is likely to transcend to other stakeholders, through discussions and collaborative projects, thus broadening the reach of the challenge.

Riege & Lindsay explained that a key challenge for collaborating with multiple stakeholders in a knowledge engagement process is the differences in priorities and underlying values of various stakeholders in their discussion of various theories in managing knowledge in public sectors [44]. This may result in a conflict of perspectives and governance. Many collaborative stakeholders in the knowledge hubs analyzed in our study were leaders in their respective fields. Discussions surrounding leadership, governance, and authority arose in various knowledge hubs around the world [26,31,32]. Furthermore, issues with role clarity, due to conflicts with governance, also arose in various knowledge hubs, which led to decreased productivity and/or unawareness of individual responsibilities in certain situations [6,32]. Unlike a commercial partnership, collaboration for knowledge engagement may not always demand clear and equal mutual benefits and reciprocity to all stakeholders, especially in multifaceted and complex issues, Khalaf explained in their study [44,45].

Funding is another key challenge for starting up and/or sustenance of a hub, as evident in the articles included in this study [26,41]. Rycroft-Malone et al. in their reflection on collaborative knowledge production pointed out that the co-creation of knowledge takes time, and without the commitment from the funders and flexibility on the timeline, it is very difficult to achieve [46]. While the end goals may be to serve communities, increase knowledge mobilization, and reduce care inequities, for example, it is still crucial to consider aspects such as infrastructure, funding for infrastructure, meeting times, and other aspects related to starting a business—before hubs are able to realize their end goals [26,38]. Overall, we see fewer challenges in comparison to the benefits, thereby emphasizing the advantages of developing knowledge hubs. Indeed, challenges are expected to arise with new endeavors.

In accordance with the commonly reported across-the-board challenges, addressing challenges was also directed to the whole group of stakeholders who collaborated in knowledge hubs. For example, conflicts with diverse perspectives were identified as a pertinent and cross-collaborator challenge. To address this conflict, many studies suggested employing power-sharing dynamics and providing contextual awareness for all stakeholders or simultaneously emphasized the need to allow for open communication and to listen to other stakeholders when they communicate [26,31,32]. The analyzed studies emphasize the need for physical infrastructure with online supporting platforms, governance rooted in teamwork, and the importance of being open-minded when engaging in collaborative knowledge production processes [26,38].

A key focus of this review was to understand the involvement of the community in the knowledge hub as we focused on the CEKHs only. There have been theories, frameworks, and models on how knowledge can be mobilized, exchanged, and implemented that outline why and how communities need to be involved. For instance, Carayannis and colleagues developed the quadruple and quintuple helix models that theorize the interrelationship of researchers, industry, government, community (i.e., public and civil society), and environment in knowledge creation and implementation [47,48]. Recently, with the decolonization trend of research, there has been an increased urge for involving the community in knowledge creation and mobilization activities. By including the community as the fourth helix of their models, Carayannis and colleagues draw on the dimension of knowledge democracy that values the role of community. Though there has been an argument that community involvement in this process might narrow the community as one private actor only instead of acting as the foundation of knowledge creation and mobilization [49], this review demonstrates community benefits by ensuring their active involvement in knowledge hubs.

### Strengths and Limitations

Our study summarizes the available literature on knowledge hubs where various stakeholders, including community members, researchers, and service providers, were involved in various steps of those projects. Despite using a search strategy, which covers multidisciplinary literature databases, most of the identified studies were from the health and wellness domain. This might be due to the transdisciplinary approach common in the health and wellness sectors. From the literature and our community engagement [50,51], it is evident that there is a need for mutual, equal, and respectful collaboration between all stakeholders of knowledge hubs to attain the greatest success in knowledge production, creation, and mobilization efforts. Collaboration entails addressing the strengths and weaknesses of all members and then capitalizing on and unifying the strengths to allow knowledge production to be a strong, co-creative process [52,53]. A successful partnership extends further than allowing academics, communities, service providers, and policymakers a seat at the table—a successful collaboration entails founding the partnership and capitalizing on the unique and valuable strengths of individual stakeholders [54,55,56].

Due to the lack of prior studies on knowledge hubs, this study included a limited number of studies for synthesis. Furthermore, due to the novel and complex understanding of knowledge and knowledge mobilization, interpretations of the evidence can vary based on individual understanding [3,4]. The coding of this study was also primarily done by one individual, with revisions by other members of our research team. In addition, it is important to note that the increased benefits that emerged for academics and communities, in comparison to service providers and policymakers, could stem from the fact that the included papers were dominantly written by academic institutions; we also further focused on community-related knowledge hubs, as per our study inclusion criteria. As such, individual biases are assumed to be a part of the analysis. However, the impact of individual biases was minimized through the qualitative description approach for the coding [22,24]. This approach limits the interpretation of data to the perspective of the data source and restrains researchers to bring their own contexts, thus limiting bias. The exclusion of non-English studies may also be considered a limitation as some knowledge hub development studies may have been missed. What is unique to this study, however, is that it fills a gap in the literature regarding the analysis and identification of strategies across knowledge hubs on how to address challenges specific to hub development.

## 5. Conclusions

This review identifies the community-engaged knowledge hubs reported in the literature and synthesizes the benefits of knowledge hubs for all stakeholders involved. In addition, the study identifies the challenges in the development and sustenance of such knowledge hubs reported by those studies as well as the potential measures to address those challenges. The benefits of knowledge hubs include the improvement of research outcomes, better knowledge mobilization and dissemination, and increased community benefits through the capacity building of community partners and their greater involvement in knowledge production and mobilization activities. There were certain challenges that include differences in priorities across sectors, limited capacities, resistance to change, and lack of funding to sustain a hub. The potential measures to overcome these challenges include taking innovative engagement efforts from the outset, provisions for in-kind contributions, and seeking multi-sectoral support. This study will aid academics, community partners/practitioners, and policymakers who intend to develop or be part of knowledge hubs with a foundational understanding of the know-how of a meaningful community-engaged knowledge hub.

This study reviews existing literature on knowledge hub development and summarizes important aspects of knowledge hub development that are likely to be useful for developing future knowledge hubs. While collaboration with the community is essential for the creation of the knowledge hub, the community’s active contribution to knowledge hubs is rarely given much attention. Academics and researchers can contribute to community-engaged research through greater exposure to perspectives of actual knowledge users as well as the inclusion of and collaboration with community members, service providers, and policymakers. There is a need to collaborate with community members, policymakers, and service providers to include a diverse perspective of actual knowledge users before initiating community interventions. Therefore, further research should focus on how collaboration with community members, policymakers, and service providers can be maintained to develop a CEKH. The findings of this review will be helpful for all stakeholders to focus on creating a CEKH that benefits all.

## Figures and Tables

**Figure 1 ijerph-20-01160-f001:**
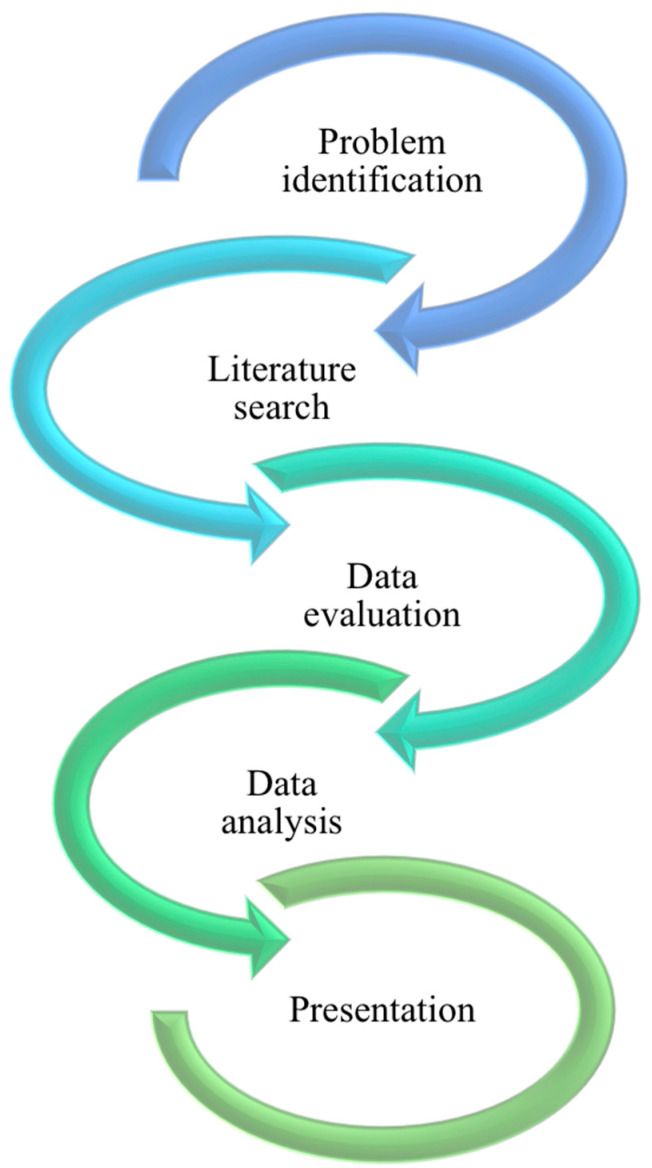
The steps of the integrative literature review.

**Figure 2 ijerph-20-01160-f002:**
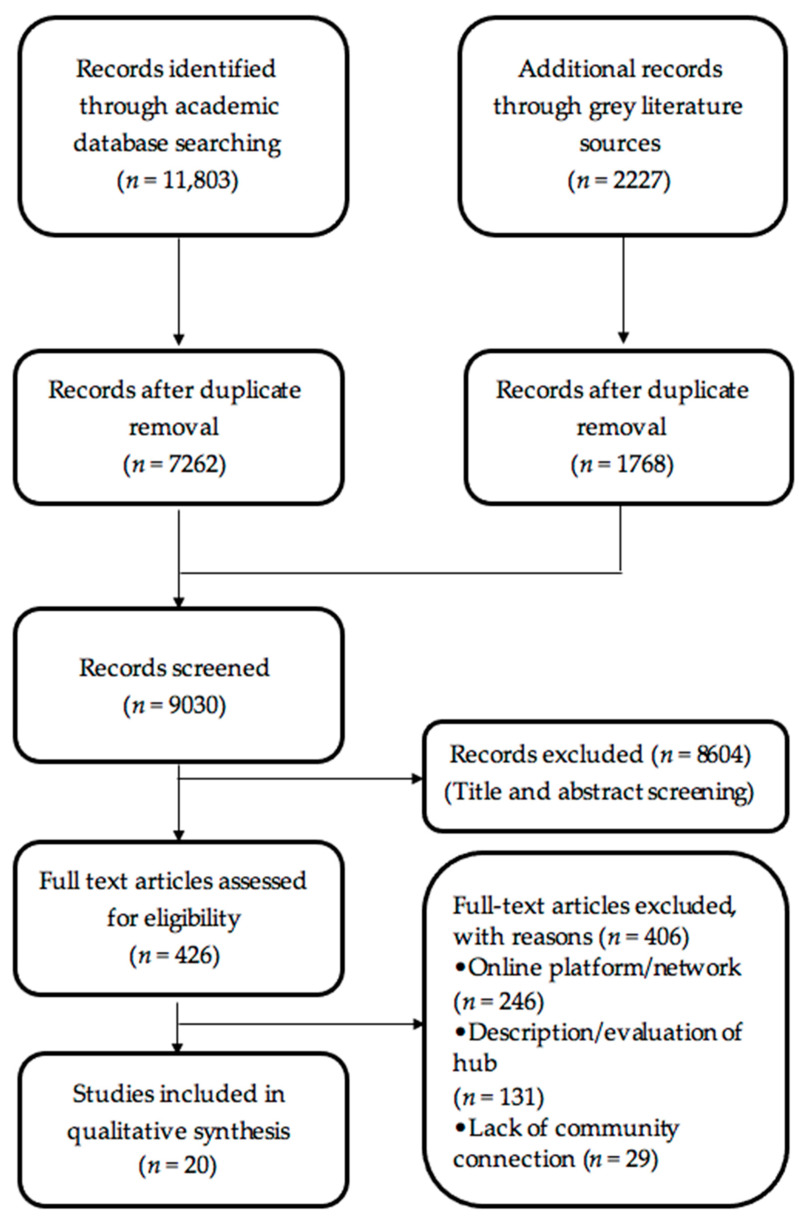
Preferred Reporting Items for Systematic Reviews and Meta-Analyses flowchart to track included and excluded study numbers.

**Figure 3 ijerph-20-01160-f003:**
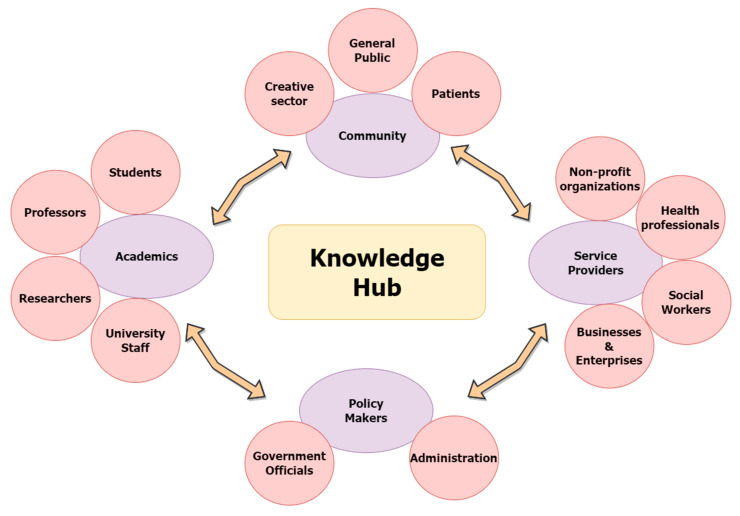
Stakeholders collaborating in knowledge hubs.

**Table 1 ijerph-20-01160-t001:** Databases searched for studies.

Database Type	Database
A. Health and wellness science	MEDLINE (Ovid) Embase CINAHL
B. Social Sciences and Humanities	4. Sociological Abstracts 5. SocINDEX with full text
C. Education	6. Education Research Complete 7. ERIC
D. Multidisciplinary	8. Academic Search Complete 9. Scopus 10. Web of Science
E. Grey literature	11. Google Scholar 12. OAISter 13. OpenGrey 14. ProQuest (theses and dissertations)

**Table 2 ijerph-20-01160-t002:** Search terms in databases.

Search Terms for Knowledge Mobilization
“knowledge translation” [keyword], “knowledge mobilization” [keyword], “knowledge exchange” [keyword], “knowledge management” [keyword; MeSH], “evidence-based practice” [keyword; MeSH], “evidence-informed practice” [keyword], “information dissemination” [keyword; MeSH], “knowledge utilization” [keyword], “knowledge transfer” [keyword], “implementation research” [keyword], “research utilization” [keyword], “Translational Medical Research” [keyword; MeSH], “diffusion of innovation” [keyword; MeSH], “Professional Practice” [keyword], “knowledge uptake” [keyword], “knowledge to action” [keyword], “knowledge integration” [keyword], “knowledge implementation” [keyword], “knowledge dissemination” [keyword], “knowledge adoption” [keyword]
**Keywords for Hub**
hub [keyword], network [keyword], lab [keyword], group [keyword], repository [keyword], platform [keyword], centre [keyword], center [keyword], system [keyword], forum [keyword], team [keyword], cluster [keyword], collaboration [keyword], core [keyword], nucleus [keyword], “focal point” [keyword], “test center” [keyword], “test centre” [keyword]

**Table 3 ijerph-20-01160-t003:** Inclusion and exclusion criteria.

Inclusion Criteria	Exclusion Criteria
1. Original research paper 2. Development of a physical knowledge hub 3. Community collaboration in the development of a knowledge hub	1. Not in English 2. Knowledge management-focused paper regarding physical/electronic management of data 3. Evidence-based practice and/or knowledge translation only within one group of researchers, nurses, physicians, etc. 4. Online platform or network 5. Lack of community involvement

**Table 4 ijerph-20-01160-t004:** Study characteristics.

Study/ Location	Study Objective	Methods	Name of the KM Hub	Objective of the KM Hub	Focus of the KM Hub	Theoretical Basis	Who Collaborated?	Who Initiated the Hub?
Michalak et al., 2012 [26]; British Columbia, Canada	To describe a cross-sectoral network for studying psychosocial factors in bipolar disorder	Case study	Collaborative RESearch Team to study psychosocial factors in Bipolar Disorder (CREST.BD)	Fundamental research and knowledge exchange on bipolar disorder	Bipolar disorder	Community-based participatory research (CBPR)	Researchers, people with bipolar disorder, their family members and supporters, health care providers	Researchers
Chambers et al., 2010 [25]; Ontario, Canada	To describe the design and activities of the Ontario Seniors Health Research Transfer Network (SHRTN)	Case study	The Ontario Seniors Health Research Transfer Network (SHRTN)	Improve the health of older adults through increasing the knowledge capacity of community care agencies and long-term care homes	Senior’s health	Knowledge transfer and network theory	Committed communities of practice members, a library service, a network of 7 research institutes, and local implementation teams	N/R
Winstein et al., 2007 [27]; USA	To describe the vision, methods, and implementation strategies used in building the infrastructure for PTClinResNet	Summary of multiple randomized Controlled Trials	The Physical Therapy Clinical Research Network (PTClinResNet)	To assess outcomes for health-related mobility associated with evidence-based physical therapy interventions across and within four different disabilities	Occupational health	N/R	Interdisciplinary clinicians–researchers and academic institutions	N/R
Wölwer et al., 2003 [29]; Germany	To describe the aims and structure of the German Research Network on Schizophrenia (GRNS) including cost-benefit aspects	Case study	The German Research Network On Schizophrenia (GRNS)	To create scientific precondition and for the implementation of preventive strategies and for the optimization of treatment and rehabilitation	Schizophrenia	N/R	Scientific Advisory Board, Executive committee, coordinators of subnetworks, principal investigators, and network members	Researchers
Wölwer et al., 2006 [30]; Germany	To describe the concept of German Research Network on Schizophrenia and the first results of studies as examples of this effort	Case study	The German Research Network On Schizophrenia (GRNS)	To create scientific preconditions for the implementation of preventive strategies and the optimization of treatment and rehabilitation	Schizophrenia	N/R	Scientific Advisory Board, Executive committee, coordinators of subnetworks, principal investigators, and network members	Researchers
Brookman-Frazee et al., 2012 [31]; USA	To describe the formation and initial outcomes of a research-community collaborative group that was developed based on community-based participatory research principles	Survey	Southern California BRIDGE Collaborative	To develop a community-wide, sustainable plan for serving infants/toddlers at risk for ASD	Autism Spectrum Disorder	Community-based participatory research (CBPR)	Transdisciplinary team of practitioners, funding agency representatives, researchers, and families of children with ASD	Collaborative effort between researchers, community providers, and early intervention funding agency representatives
Abdul-Adil et al., 2010 [32]; Chicago, USA	To bring empirically based practice to the ‘‘real world’’, front-line practice settings of community-based agencies	Case study	University-Community Mental Health Center Collaboration	To develop effective and accessible interventions	Mental health	Knowledge exchange	Researchers and service providers	N/R
Brekke et al., 2009 [33]; USA	To describe the principle and model of Interventions and Practice Research Infrastructure Program (IPRISP)	Case study	The Interventions and Practice Research Infrastructure Program (IPRISP)	To establish the evidence base for improving mental health care available in diverse communities	Mental health	Implementation research	Universities and community-based agencies	Policymakers (National Institute of Mental Health [NIMH])
Manns et al., 2014 [34]; Canada	To describe the Canadian Kidney Knowledge Translation and Generation Network (CANN-NET) established to improve care of the kidney patients	Case study	Canadian Kidney Knowledge Translation and Generation Network (CANN-NET)	Establishing a national knowledge translation and generation network in kidney disease	Kidney diseases	Knowledge Translation	Canadian Society of Nephrology (CSN) Clinical Practice Guidelines Group, kidney researchers, decision-makers, and knowledge user	Canadian stakeholders in kidney disease
Edwards et al., 2016 [35]; Jamaica, Kenya, South Africa, and Uganda	To determine the impact of establishing leadership hubs on HIV care by nurses	Quasi-experimental	Leadership hubs on the uptake of evidence-informed nursing practices and workplace policies for HIV care	To improve policies and patient care within health care institutions orchestrated through a district-level change process	HIV	Community-based participatory research (CBPR)	Nurses, midwives, nurse managers, nurse researchers, decision-makers from the Ministry of Health, local representatives from nursing or other health professional and regulatory bodies and unions, community representatives from community groups active on HIV issues (e.g., people living with HIV), person(s) living with HIV or AIDS	N/R
Sol et al., 2018 [41]; The Netherlands	To find out factors that foster (un)successful social learning in governance sustainability transitions networks	Case study	Duurzaam Door (Moving Forward Sustainably)	To develop equal partnerships, towards, for example, local/regional energy cooperatives	Sustainability	N/R	Municipalities, entrepreneurs, educational institutes, NGO’s, citizens, and other actors	N/R
Steens et al., 2018 [40]; Netherlands	To elaborate on the origins, goals, and conceptual framework of the Academic Collaborative Centre (ACC), and describe the implementation process	Case study	Building an Academic Collaborative Centre (ACC)	To build knowledge about enabling and restricting mechanisms that can explain the impact of social interventions and to facilitate the formation of a learning organization where critical questions are raised, different ways of understanding social reality are combined and stakeholders work collaboratively towards a better quality of care	Social work	N/R	Researchers, the social service agency management, social workers, and service users	Researchers
Shrager et al., 2010 [43]; California, USA	This paper reports on a series of experiments in this area and a prototype implementation using a research platform called CACHE-BC	Narrative review	Collaborative analysis of Competing Hypotheses Bayes Community (CACHE-BC	To support experimentation with different structures of interaction between individual and community cognition in science and to enable us to prototype computational support for those structures	Open science	N/R	N/R	Researchers
Salem et al., 2005 [38]; Chicago, USA	To describe Chicago’s approach to developing and supporting a network of linked coalitions engaged in Mobilizing for Action through Planning and Partnerships (MAPP) processes in five diverse communities and its outcomes	Case study	Mobilizing for Action through Planning and Partnerships (MAPP)	To increase community capacity, build new partnerships, provide coalitions with access to decision-makers, and inform the role of local public health agencies	Public health systems	Community engagement	Chicago Department of Public Health staff and ad hoc committees of select community leaders, key community stakeholders, neighborhood organizations and residents	Policymakers (Chicago Department of Public Health)
Moreton 2016 [42]; UK	To collaborative work between arts and humanities disciplines in UK Higher Education and the creative economy	Case study	Knowledge Exchange Hubs for the Creative Economy	To exchange knowledge and support collaborative work	Creative economy	Knowledge exchange	Higher education institutions, creative sector, microbusinesses, small and medium-sized enterprises (SMEs), and Arts and Humanities researchers	Researchers (UK Arts and Humanities Research Council)
Fitzgerald and Harvey 2015 [6]; England, UK	To explore the relationship between organizational form and the function(s) of a translational network	Analysis of archival and documentary data	Collaboration for Leadership in Applied Health Research and Care (CLAHRC)	To develop an innovative and distributed model for conducting applied health research and translating research findings into improved outcomes for patients by connecting practitioners	Health care	Knowledge translation and mobilization	University and multiple National Health Service (NHS) organizations	N/R
Campbell et al., 2017 [39]; Ontario, Canada	To discuss approaches for the governance and implementation of a system-wide knowledge mobilization network and the strategies, challenges, and successes of knowledge mobilization projects	Content analysis	Knowledge Network for Applied Education Research (KN/AER)	To support the adaptation and implementation of evidence from research and professional knowledge to inform changes in educational practices	Education	Knowledge mobilization	Universities and Ontario Ministry of Education	Researchers and policymakers (a tripartite initiative in Canada involving the Ontario Ministry of Education, University of Toronto, and Western University)
Kramer and Wells 2005 [28]; Ontario, Canada	To offer an overview and an evaluation of the process of transferring a complex body of knowledge from a research institute to workplace parties	Content analysis	Building Networks to Facilitate Knowledge Transfer	To transfer knowledge about workplace safety/ergonomics to a group of practitioner-based associations	Occupational health	Knowledge transfer and network theory	Consultants and ergonomists from multiple Health and Safety Prevention systems	Researchers (Institute for Work & Health)
Phillips et al., 2019 [37]; Indiana, USA	Describes the process and outcomes of an academic–practice partnership facilitated by nurse educators in both academic and practice settings	Case study	Academic and practice partnership: A win-win	To provide a foundational structure to help nurses achieve educational and career advancement, prepare students for future to practice, provide mechanisms for lifelong learning, and provide a structure for nurse residency programs	Nursing	N/R	Students, nurses, and nurse leaders (academia and practice)	Service providers
Baumbusch et al., 2008 [36]; British Columbia, Canada	How participatory approach to knowledge translation developed during an ongoing program of research concerning equitable care for diverse populations	Case study	Knowledge Translation between Research and Practice in Clinical Settings	To break down traditional barriers between researchers and practitioners by promoting a shift away from the typical roles of each in the research process.	Healthcare	Knowledge Translation	Researchers and practitioners	Researchers

**Table 5 ijerph-20-01160-t005:** Benefits of and challenges for developing knowledge hubs.

Benefits		
Stakeholder	Themes	Benefits of Developing Knowledge Hubs (*n*)
Academic	Improved dissemination process	Increasing information dissemination (2)
		Faster dissemination of research findings (1)
	Improving image of academia	Academic accountability (3)
		Breaking stereotypes (1)
		Increased trust of the community in research (1)
	Knowledge sharing	Facilitating cross-hub knowledge exchange (4)
		Mutual respect for each other’s knowledge (1)
	Research	Increased productivity (3)
		Increased research opportunities (5)
		Reducing bias in research (1)
		Standardized research procedures (7)
		Practice relevant research (7)
		Practitioners involved in the research process (1)
		Collaboration in the research analysis (1)
		Collaboration in the research process (5)
		Divide and conquer research (1)
		Increased participant-recruitment potential (1)
		Community-based participatory research (1)
Community	Appropriate community interventions	Community intervention agreement (1)
		Community-need sensitive interventions (18)
		Expert informed interventions (1)
		Feasible interventions (1)
		Flexibility in implementing EBP (1)
		Early intervention (2)
		Collaboration to determine appropriate intervention (2)
	Collaboration with the community in research	Community-based participatory research (1)
		Practice related research (7)
		Increased trust of the community in research (1)
		Commitment to the community (1)
		Power sharing with the community (1)
		Shared decision-making (6)
	General community benefits	Decreased environmental impact (1)
		Increased community participation (1)
		Reduce community food insecurity (3)
		Using research to increase issue understanding (1)
	Improved care	Evidence-based medicine (EBM) informed care (5)
		Improved community care (14)
		Reduce care inequities (4)
		Increased access to care (5)
Service providers	Improved care provision	Improved assessment instruments (1)
		EBM informed care (5)
		Improve community care (14)
	Research	Practitioners involved in the research process (1)
		Using research to inform practices (9)
Policymakers	Knowledge translation	Using research to justify action (2)
		Using research to inform policy (8)
		Using research to increase issue understanding (1)
	Policy assessment	Improved assessment instruments (1)
	Policy implementation	Improving policy implementation (3)
Generalized benefits for all actors involved with the hub	Increased capacity	Personal capacity building (10)
		Sharing capacity (6)
		Increased resources accessibility (4)
		New collaborations (6)
		Adoption of knowledge (2)
		Sharing of experience-based knowledge (2)
		Knowledge exchange (16)
		Knowledge translation (6)
		Knowledge mobilization (19)
	Knowledge production	Knowledge production (20)
		Increased productivity (3)
	Networking	Cross network accessibility through websites (2)
		Horizontal networking (3)
		International linkages (1)
		Network building through social media (5)
		Vertical networking (3)
	Other	Generalizability of implementation possibility (1)
		Hub sustained over time (5)
		Mutual benefit of collaboration (3)
**Challenges**		
Academics	Challenges with new ideas	Challenges in implementing a theoretical model (2)
		Challenges in reframing research (1)
		Confusion with new knowledge (2)
	Collaborative resistance	Resistance to change (2)
		Problems with shared governance (1)
Community	Lack of evaluatory means	Lack of evaluation of community impact (1)
		Lack of evaluation of research-community partnership (1)
	Poor trust in research/academia	Declining trust (1)
		Lack of community trust in interventions (1)
		Preconceived notions of hubs (1)
	Governance structure	Separation between senior and junior authority (1)
	Lack of prior knowledge/training	Lack of knowledge relevance (1)
Service providers	Time constraints	Lack of capacity to allocate dedicated time (6)
	Skill set limitation	Lack of resources/skilled personnel to support hub specific activities (1)
	Sustainability	Challenge in maintaining continuity (1)
Policymakers	Conflicts with external systems	Challenges validating hub in the healthcare system (6)
		Socio-political challenges (1)
All stakeholders	Diverse stakeholders vs. hub interest	Challenges with intellectual territory (1)
		Concerns with diverse collaborators (2)
		Consensus challenges (1)
		Lack of mutual understanding (3)
		Scheduling challenges (1)
		Stakeholder vs. hub interest (1)
	Starting up a new hub	Geographical limitations (1)
		Limited funding (9)
		Recruitment challenges (1)
		Technological challenges (1)
	Sustainability	Challenge in maintaining continuity (1)
		Hub sustainability challenges (2)
	Overall governance structure	Challenges due to lack of administration (1)
		Challenges in role organization (2)
		Lack of responsibility clarity (1)
		Lack of role clarity (1)
	Lack of time and resources	Short developmental timeframe (1)
		Time-consuming (6)
	Knowledge communication and network connections	Challenges with knowledge translation (1)
		Delay in information dissemination (2)
		Lack of intra-hub communication (3)
		Lack of replication across hubs (2)
		Challenge in network connection (3)

**Table 6 ijerph-20-01160-t006:** Addressing the challenges of developing knowledge hubs.

Addressing the Challenges	
Themes	Strategies/Factors Important for Addressing Challenges Presented during Knowledge Hub Development *(n*)
Addressing start-up challenges	Be creative to attain external funding (5)
	Pooling in-kind contributions from the members (1)
	Have stakeholders be open to new and diverse ideas (1)
	Need for infrastructure and a physical meeting space (1)
	Need for organizational system (5)
Need for external involvement for sustainability	Importance of establishing a hub in healthcare systems through collaboration with healthcare system administration (1)
	Pooling in-kind contributions from the members (1)
	Need for government collaboration and support (2)
Addressing individual stakeholder issues	Need for stakeholder flexibility in adapting to others’ ideas (3)
	Need for short-term endurance and perseverance to achieve long-term hub goals (1)
	Need to listen to all stakeholders and work cohesively rather than individually (1)
	Willingness to work towards consensus to sustain collaboration (3)
	Need for communication as soon as conflict is identified (3)
	Need for power sharing during decision-making (2)
	Identify individual stakeholder’s strengths and weaknesses to form a strength-based approach (1)
	Addressing dissemination delay through the use of online platforms and social media (1)
Other	Adapt to theoretical implementation challenges through education of all members (6)

## Data Availability

The details of the articles derived from academic and grey literature search are provided in an online data repository, figshare [19,20].

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
