# Peer review of "The Benefits, Challenges, and Strategies toward Establishing a Community-Engaged Knowledge Hub: An Integrative Review"

_ijerph, 2023, doi:10.3390/ijerph20021160_

Round 1

Reviewer 1 Report (Previous Reviewer 1)

All my previous comments have been addressed, the paper is ready to be published.

My only proposal is that they could upload the data filtered in the study (table 4) as a dataset in a public website (like figshare, Zenodo or similar) and include the URL/DOI in the paper. This way other authors could use it for future research (and paper can be aligned with knowledge hub intentions ;)

Congratulations to the authors

Author Response

We really appreciate your contribution to improving our manuscript.

Thank you so much for this excellent suggestion, which we never thought about before. From now on, we will be sharing our review related data in your suggested way to keep our efforts aligned with the knowledge hub intentions.

We have uploaded the academic and grey literature datasets created for our study in figshare and mentioned about that in our revised manuscript. Please see page 5, lines 157-158.

Reviewer 2 Report (New Reviewer)

Although the paper attempts to provide the readers with the comprehensive analysis of the existing literature on community-engaged knowledge hubs seemingly using sophisticated research design, the actual scientific contribution and value-added of the study are rather limited.

Apart from making the literature review what is the value-added of the paper?

The Conclusions are rather trivial and do not contribute to the state of art.

The paper in current form needs significant improvement and the clear presentation of the contribution of the results of the conducted analysis in the paper to the relevant literature.

Although the Authors start with impressive sample of more than 11,000 papers, the conclusions of the study are based on the subjective analysis of merely 20 papers (¾ of which are focused on healthcare). 

Instead of providing such an extensive literature review, which in fact does not offer any valuable scientific contribution, maybe the Authors should focus directly on the healthcare sector using their own experience to improve the value added of the study to make it more appealing to the journal’s readers. 

The concept of an inevitable role of collaboration and interactions among the actors of the knowledge creation, diffusion, and absorption has been widely discussed in economic literature. Therefore, it is strongly advisable that the Authors compare and highlight the differences between the concept of knowledge hubs and analogical ones, like for example the quadruple or quintuple innovation helix frameworks that describe academic-industry-government-community-environment interactions in a knowledge-based economy.

In lines 72-88 the Authors refer to the establishment and actions taken by NRN including the results of their own research in the area of community-based knowledge hubs. Before presenting those results the Authors should, however, explicitly indicate their participation in the NRN initiative.

As the Authors admit, the focus of 15 out of 20 knowledge hubs was related to health and wellness (other knowledge hubs were on education, social work, sustainability, creative economy, and science in general, particularly open science concept). However, Figure 3. describing stakeholders collaborating in knowledge hubs presents only service providers limited to the healthcare sector: i.e. allied health professionals, nurses, social workers and physicians. Moreover, in the final part of the paper, in general Conclusions, the Authors state that: “The benefits of knowledge hubs include the improvement of health outcomes through the capacity building of service providers and their greater involvement in knowledge production and mobilization activities.”

As regards the list of keywords provided by the Authors the term ‘racialized community’ doesn’t seem to be justified by the actual content of the paper. 

The Introduction section should end with presentation of the paper's structure.

In lines 156-159 two Authors are indicated by initials whereas the third one is not. 

Although the quality of English language in the paper is generally fine, the manuscript needs some further proofreading with respect to grammar, wording and style (e.g. line 42 – ‘…are seldom collaborated…’, line 140 – ‘leterature’, line 37 – ‘emerging’, line 286 – ‘studies that collaborated’). Reference style needs standardization – see e.g. line 347.

Author Response

Although the paper attempts to provide the readers with the comprehensive analysis of the existing literature on community-engaged knowledge hubs seemingly using sophisticated research design, the actual scientific contribution and value-added of the study are rather limited. Apart from making the literature review what is the value-added of the paper?

Reply:

Thank you for your comment. As you have already mentioned, we have conducted a comprehensive literature review to summarize the existing literature on community-engaged knowledge hubs. We undertook this review as a part of our implementation research to explore what has been done so far to seek to understand what worked and what did not work in previous knowledge hub initiatives. Implementation research particularly focuses on challenges of employing a concept within real-world conditions and not entirely the production of scientific knowledge. We intended to summarize the challenges and possible ways to overcome those challenges so that academics, community partners/practitioners, and policymakers who intend to develop or be part of knowledge hubs are aware of the issues they need to keep in mind.

The Conclusions are rather trivial and do not contribute to the state of art. The paper in current form needs significant improvement and the clear presentation of the contribution of the results of the conducted analysis in the paper to the relevant literature.

Reply:

Thank you for your comment. Based on your comment, we have modified the conclusion section in the revised manuscript for better clarity.

Please see pages 19 lines 418-432

“This review identifies the community-engaged knowledge hubs reported in the literature and synthesizes the benefits of knowledge hubs for all stakeholders involved. In addition, the study identifies the challenges to develop and sustain such knowledge hubs reported by those studies as well as the potential measures to address those challenges. The benefits of knowledge hubs include the improvement of research outcomes, better knowledge mobilization and dissemination, and increased community benefits through the capacity building of community partners and their greater involvement in knowledge production and mobilization activities. There were certain challenges that include differences in priorities across sectors, limited capacities, resistance to change, and lack of funding to sustain a hub. The potential measures to overcome these challenges include taking innovative engagement efforts from the outset, provisions for in-kind contributions, and seeking multi-sectoral support. This study will aid academics, community partners/practitioners, and policymakers who intend to develop or be part of knowledge hubs with a foundational understanding of know-how of a meaningful community-engaged knowledge hub.”

Although the Authors start with impressive sample of more than 11,000 papers, the conclusions of the study are based on the subjective analysis of merely 20 papers (¾ of which are focused on healthcare). 

Reply:

Thank you for your comment. Based on our study objective and inclusion/exclusion criteria (Table 3) we employed two-steps (title-abstract and full-text) screening process by two-independent-reviewers. This yielded 20 papers for the synthesis (Figure 2). Indeed, 3/4th of them focused on healthcare. We would like to bring to your kind attention that, we did not limit to health or any particular sector as a part of our study design. We encountered studies that only created a knowledge hub among researchers or among practitioners, without any community involvement. Those were excluded due to not fitting to our criteria as our objective was to include the knowledge hub associated with communities.

Despite using an open search strategy and searching multidisciplinary literature databases (Table 1), most of the studies were related to health and wellness. This may be due to the increasing trend of community engaged knowledge mobilization approach in the health and wellness sectors.

We also acknowledged this in the discussion section.

See page 18, lines 388-391

“Despite using a search strategy, which covers multidisciplinary literature databases, most of the identified studies were from the health and wellness domain. This might be due to the transdisciplinary approach common in the health and wellness sectors.”

Instead of providing such an extensive literature review, which in fact does not offer any valuable scientific contribution, maybe the Authors should focus directly on the healthcare sector using their own experience to improve the value added of the study to make it more appealing to the journal’s readers. 

Reply:

Thank you for such an excellent suggestion to develop an article based on our experiences. Definitely, we will write such an article in future. Our understanding gained from this extensive literature review will aid us in establishing an impactful community-engaged knowledge hub based on a good understanding of potential challenges and solutions. We believe that this study will aid academics, community partners/practitioners, and policymakers who intend to develop or be part of knowledge hubs with a foundational understanding of know-how of a meaningful community-engaged knowledge hub.

The concept of an inevitable role of collaboration and interactions among the actors of the knowledge creation, diffusion, and absorption has been widely discussed in economic literature. Therefore, it is strongly advisable that the Authors compare and highlight the differences between the concept of knowledge hubs and analogical ones, like for example the quadruple or quintuple innovation helix frameworks that describe academic-industry-government-community-environment interactions in a knowledge-based economy.

Reply:

Thank you for suggesting this point. Based on your comment, we have added the following in our discussion section. Please see page 19, lines 370-383.

“A key focus of this review was to understand the involvement of the community in the knowledge hub as we focused on the CEKHs only. There have been theories, frameworks, and models on how knowledge can be mobilized, exchanged, and implemented that outline why and how communities need to be involved. For instance, Carayannis and colleagues developed the quadruple and quintuple helix models that theorize the interrelationship of researchers, industry, government, community (i.e., public and civil society), and environment in knowledge creation and implementation [47,48]. Recently, with the decolonization trend of research, there has been an increased urge for involving the community in knowledge creation and mobilization activities. The inclusion of the community as the fourth helix of their models Carayannis and colleagues draw on the dimension of knowledge democracy that values the role of community. Although, there has been an argument that community involvement in this process might narrow the community as one private actor only instead of acting as the foundation of knowledge creation and mobilization [49], this review demonstrates community benefits by ensuring their active involvement in knowledge hub.”

In lines 72-88 the Authors refer to the establishment and actions taken by NRN including the results of their own research in the area of community-based knowledge hubs. Before presenting those results the Authors should, however, explicitly indicate their participation in the NRN initiative.

Reply:

Thank you for your comment.

We have indicated explicitly the authors’ participation in NRN.

See page 2, lines 76-77

Also, for your kind note is that NRN is already included as authors’ affiliation for all the authors.

As the Authors admit, the focus of 15 out of 20 knowledge hubs was related to health and wellness (other knowledge hubs were on education, social work, sustainability, creative economy, and science in general, particularly open science concept). However, Figure 3. describing stakeholders collaborating in knowledge hubs presents only service providers limited to the healthcare sector: i.e. allied health professionals, nurses, social workers and physicians.

Reply:

Thank you for pointing out the ambiguity. We have updated the figure to include those missing sectors (See page 8, figure 3).

Moreover, in the final part of the paper, in general Conclusions, the Authors state that: “The benefits of knowledge hubs include the improvement of health outcomes through the capacity building of service providers and their greater involvement in knowledge production and mobilization activities.”

Reply:

Thank you for you comment. We have updated the conclusion section based on your previous comments, which takes care of this comment as well. Pages 19, lines 418-432

As regards the list of keywords provided by the Authors the term ‘racialized community’ doesn’t seem to be justified by the actual content of the paper. 

Reply:

Thanks for pointing that out. We have removed the keyword and added a few more relevant keywords.

The Introduction section should end with presentation of the paper's structure.

Reply:

Thanks for your comment. We provided the paper’s structure at the end of the introduction. Please see page 3, lines 114-118

“The objectives of this study are to analyze and summarize existing literature on various kinds of knowledge hubs that collaborated with the community in some way. This systematic integrative review draws on the existing theoretical and empirical studies and describes the benefits, challenges, and ways to mitigate challenges for the development of a wide range of knowledge hubs.”

In lines 156-159 two Authors are indicated by initials whereas the third one is not. 

Reply:

Thank you for pointing this out. We have added the initial of the third reviewer (TCT) in that line.

Although the quality of English language in the paper is generally fine, the manuscript needs some further proofreading with respect to grammar, wording and style (e.g. line 42 – ‘…are seldom collaborated…’,

line 140 – ‘leterature’, line 37 – ‘emerging’, line 286 – ‘studies that collaborated’). Reference style needs standardization – see e.g. line 347.

Reply:

Thank you for highlighting them. We have corrected them in the revised manuscript. Also, we have corrected the reference style across the article.

This manuscript is a resubmission of an earlier submission. The following is a list of the peer review reports and author responses from that submission.

Round 1

Reviewer 1 Report

Authors make a systematic research to identify existing primary literature on knowledge hubs, identifying the benefits and challenges

Authors have make a nice work, and with minor adjustements can be published:

* Abstract. There are some very vague (without information) phrases, like "We identified admissible studies based on predetermined inclusion criteria". Authors could better include more especific and significant information, like some figures of the study

* References do not follow usual font, and are without brackets

* Section 2.1 would be better to understand if a diagram of the process is included

* There is a box in Figure 1 that is too small and the text "(Title and abstract screening)" is partially hidden. Additionally, some arrow lines are not properly set to the pointed boxes

* I cannot read the heading of table 4: black font and black cell background

* Authors conclude that must have are related to health. Due to the limited amount of results (just 20), they could discuss if they came from the same databases (form example, those specialized in "Health and wellness science") and comment if there is (or not) a lack papers about hubs in other databases

* conclusions are too short, and not descriptive at all. Phrases like "This study identifies considerable benefits of knowledge hubs for all stakeholders involved" are not suitable as a conclusion. Authors should brief those benefits, taken from the dicussion sections (and the same for all findings)

There are some English errors:
The focus of the majority of these knowledge mobilization hubs were

In a lots

Author Response

Comment
Authors make a systematic research to identify existing primary literature on knowledge hubs, identifying the benefits and challenges.
Authors have make a nice work, and with minor adjustements can be published:

Reply:

Thank you for your positive comments.

Comment
* Abstract. There are some very vague (without information) phrases, like "We identified admissible studies based on predetermined inclusion criteria". Authors could better include more especific and significant information, like some figures of the study

Reply:
Thank you for your comment. We have rewritten the sentence using the appropriate information.

Please see page 1, lines 22-24

“Following a comprehensive search of academic and grey literature databases, we screened 9,030 unique articles using predetermined inclusion criteria and identified 20 studies for final synthesis.”

Comment
* References do not follow usual font, and are without brackets

Reply:

Thank you for your comment. The version we worked on is a formatted version from the journal. So we kept the references according to the journal formatting editors. 

Comment
* Section 2.1 would be better to understand if a diagram of the process is included

Reply:

Thank you for your constructive comments and suggestions. We have added a diagram of the process in section 2.1, page 3.

Comment
* There is a box in Figure 1 that is too small and the text "(Title and abstract screening)" is partially hidden. Additionally, some arrow lines are not properly set to the pointed boxes
* I cannot read the heading of table 4: black font and black cell background

Reply:

Thank you for your comment. We have addressed this concern in the revised manuscript.

Please see pages 6 and 8.

Comment
* Authors conclude that must have are related to health. Due to the limited amount of results (just 20), they could discuss if they came from the same databases (for example, those specialized in "Health and wellness science") and comment if there is (or not) a lack papers about hubs in other databases

Reply:

Thank you for your constructive comment. We have commented on this concern in the discussion section of our revised manuscript.

Please see page 17, lines 348-350.

“Despite using a search strategy which covers multidisciplinary literature databases, most of the identified studies were from the health and wellness domain. This might be due to the transdisciplinary approach taken in the health and wellness sectors.”

Comment
* conclusions are too short, and not descriptive at all. Phrases like "This study identifies considerable benefits of knowledge hubs for all stakeholders involved" are not suitable as a conclusion. Authors should brief those benefits, taken from the dicussion sections (and the same for all findings)

Reply:

Thank you for your constructive comment. Based on your comment and the comment from reviewer 2 we have expanded our conclusion section to dwell on further research development and possible application.

Please see page 18 and lines 386-390:

“The benefits of knowledge hubs include the improvement of health outcomes through the capacity building of service providers and their greater involvement in knowledge production and mobilization activities. Therefore, despite the challenges in knowledge hub development highlighted in this study, knowledge hub presents benefits for various stakeholders, including academics, service providers, community members, and policymakers.”

and See page 19, lines 396-405:

“Academics and researchers can contribute to community-engaged research through greater exposure to perspectives of actual knowledge users, the inclusion of and collaboration with community members, service providers and policymakers. There is a need to collaborate with community members, policymakers, and service providers to include a diverse perspective of actual knowledge users before initiating community interventions. Therefore, further research should focus on how collaboration with community members, policymakers, and service providers can be maintained to develop a CEKH. The findings of this review will be helpful for all stakeholders to focus on CEKH creation that benefits all.”

Comment
There are some English errors:
The focus of the majority of these knowledge mobilization hubs were

In a lots

Reply:

Thank you for pointing out the mistake. We have corrected those errors.

Reviewer 2 Report

Dear authors,

Please dwell on importance and actuality of studying community knowledge hubs ,especially in a certain field of knowledge you are looking into.

Line 20: To summarize existing primary literature – what does it mean - existing primary literature?

Please balance line 33 and line 83:

Line 33: This study addresses a literature gap regarding comparisons of knowledge hubs and stakeholder experiences.

Line 83: Globally, literature on knowledge engagement hubs is extensive

Please provide definition of community-engaged hub or/and knowledge engagement hub and/or community knowledge hub, also in relation to Fig. 2

Line 95 The objectives of this study are to analyze and summarize existing primary research on knowledge engagement hubs. Please relate to the title key words.

The objective of research line 111 may be well served by this research.

Based on the title key words, please summarise benefits and strategies for community knowledge hub. It is not obvious from discussion and conclusion.

Please dwell on further research development or possible application.

Author Response

Dear authors,
Please dwell on importance and actuality of studying community knowledge hubs ,especially in a certain field of knowledge you are looking into. Line 20: To summarize existing primary literature – what does it mean - existing primary literature?

Reply:

Thank you for your comment. Indeed this paper is about the importance of the community knowledge hub and we have dwelled on that issue in this manuscript. By primary literature, we were referring to original articles. Based on your comment we have replaced the term “primary literature” with “original research articles”.

Please see page 1, line 19.

Comment
Please balance line 33 and line 83:
Line 33: This study addresses a literature gap regarding comparisons of knowledge hubs and stakeholder experiences.
Line 83: Globally, literature on knowledge engagement hubs is extensive

Reply:

Thank you so much for pointing out this inconsistency. We have modified line 85 of revised manuscrupt to address this issue and to match line 33.

Comment
Please provide definition of community-engaged hub or/and knowledge engagement hub and/or community knowledge hub, also in relation to Fig. 2

Reply:

Thank you so much for pointing out this issue. Based on your comment, across the whole manuscript, we have now used the term “community-engaged knowledge hub (CEKH)” consistently.

Comment
Line 95 The objectives of this study are to analyze and summarize existing primary research on knowledge engagement hubs. Please relate to the title key words.

Reply:

Thank you so much for pointing out this issue. Based on your comment, across the whole manuscript, we have now used the term “community-engaged knowledge hub (CEKH)” consistently.

Comment

The objective of research line 111 may be well served by this research.
Based on the title key words, please summarise benefits and strategies for community knowledge hub. It is not obvious from discussion and conclusion.

Reply:

Thank you for your comment. The benefits and the challenges are summarized in Table 5. The strategies related to the community-engaged knowledge hub are summarized in Table 6.

Please see pages 13-14 and page 15.

Comment
Please dwell on further research development or possible application.

Reply:

Thank you for your constructive comment. Based on your comment and the comment from reviewer 1 we have expanded our conclusion section to dwell on further research development and possible application.

Please see page 18-19 and lines 386-390:

“The benefits of knowledge hubs include the improvement of health outcomes through the capacity building of service providers and their greater involvement in knowledge production and mobilization activities. Therefore, despite the challenges in knowledge hub development highlighted in this study, knowledge hub presents benefits for various stakeholders, including academics, service providers, community members, and policymakers.”

and See page 19, lines 396-405:

“Academics and researchers can contribute to community-engaged research through greater exposure to perspectives of actual knowledge users, the inclusion of and collaboration with community members, service providers and policymakers. There is a need to collaborate with community members, policymakers, and service providers to include a diverse perspective of actual knowledge users before initiating community interventions. Therefore, further research should focus on how collaboration with community members, policymakers, and service providers can be maintained to develop a CEKH. The findings of this review will be helpful for all stakeholders to focus on CEKH creation that benefits all.”

Round 2

Reviewer 2 Report

Dear authors,

Thank you for local changes introduced. However, the research and its objective to review the current literature should have some meaningful purpose. Currently the work done does not qualify for the research itself but rather considering a tool to review the literature. There is no research novelty, practical value or research findings that would derive from using this tool.

Author Response

Reviewer 2
----------------

Comment
Dear authors,

Thank you for local changes introduced.

Reply:

We really appreciate your time and effort for reviewing our manuscript. Thank you so much.

Comment

However, the research and its objective to review the current literature should have some meaningful purpose. Currently the work done does not qualify for the research itself but rather considering a tool to review the literature. There is no research novelty, practical value or research findings that would derive from using this tool.

Reply:

In our community engaged program of research, we have recognized that the knowledge mobilization (KM) or knowledge translation (KT) with the racialized/ethnic-minority communities is not reaching its level of optimal impact. We realized that the implementation of these KT and KM efforts is the issue which we need to understand in terms of “what can be improved”. We beg to differ from your comment that this work does not have any meaningful purpose or practical value.

Also, with due respect, we disagree with you on your comment that the work does not qualify “for the research itself”. Conducting reviews is a well-established research process and has been contributing to advance knowledge.   

Base on your comment, we have added few lines to emphasis our context and the importance of this research to our context.

Please see page 2, lines 73-81